# Multifunctionalized Mesostructured Silica Nanoparticles Containing Mn2 Complex for Improved Catalase-Mimicking Activity in Water

**DOI:** 10.3390/nano12071136

**Published:** 2022-03-29

**Authors:** Tristan Pelluau, Saad Sene, Beltzane Garcia-Cirera, Belen Albela, Laurent Bonneviot, Joulia Larionova, Yannick Guari

**Affiliations:** 1ICGM, University Montpellier, CNRS, ENSCM, 34000 Montpellier, France; tristan.pelluau@umontpellier.fr (T.P.); saad.sene@umontpellier.fr (S.S.); 2Departament de Química Inorgànica i Orgànica, Universitat de Barcelona, 08028 Barcelona, Spain; beltzane.garcia@antares.qi.ub.edu; 3Laboratoire de Chimie, ENS de Lyon, Université de Lyon, 69007 Lyon, France; laurent.bonneviot@ens-lyon.fr

**Keywords:** nanoparticle, mesostructured silica, dendritic silica, manganese compounds, catalytic activity, catalase

## Abstract

We report the synthesis of a hybrid nanocatalyst obtained through the immobilization of bio-inspired [{Mn(bpy)(H_2_O)}(µ-2-MeC_6_H_4_COO)_2_(µ-O){Mn(bpy)(NO_3_)}]NO_3_ compound into functionalized, monodispersed, mesoporous silica nanoparticles. The in situ dual functionalization sol–gel strategy adopted here leads to the synthesis of raspberry-shaped silica nanoparticles of ca. 72 nm with a large open porosity with preferential localization of 1,4-pyridine within the pores and sulfobetaine zwitterion on the nanoparticles’ periphery. These nano-objects exhibit improved catalase-mimicking activity in water thanks to the encapsulation/immobilization of the catalytic active complex and high colloidal stability in water, as demonstrated through the dismutation reaction of hydrogen peroxide.

## 1. Introduction

Mesoporous silica nanoparticles (MSN) have seen an extensive development for the last several decades due to numerous advantages linked with their tunable size, morphology, important surface area and volume of pores, mesoporosity, functionalizable surface, chemical and thermal stability, and mechanical resistance and biocompatibility within certain concentration ranges [1]. MSN of different morphologies (spherical, helical fibers, tubules, crystals nanoparticles, etc.) and porosities (hexagonal, cubic, lamellar, wormhole-like, etc.) with size ranging from a few tens to several hundred nanometers have been widely designed and investigated for applications in different fields including catalysis, biology, optics, gas capture, or waste remediation [2,3,4,5,6,7,8,9].

Among different challenges related to the design of MSN with controlled size, morphology and optimized properties, the functionalization of their internal and external surfaces has attracted particular attention. Two approaches consisting of: (*i*) the direct co-condensation during the MSN synthesis, or (*ii*) the post-synthetic anchoring of functional groups, have been developed for this purpose. The choice between these two strategies is usually governed by the targeted application, MSN morphology and the preferential area of grafting [10,11]. The external surface functionalization is often used to increase the colloidal stability of MSN, because the latter suffers from a high density of surface silanol groups causing aggregation [12]. Such functionalization is also essential for biomedical applications because an appropriate surface coverage permits enhancing the blood circulation time, and ensures cells’ internalization and targeting to desired cells and tissues [1,13]. Moreover, it can also decrease biological side effects and increase bioavailability and selectivity [14,15,16]. Numerous works have reported the external surface functionalization with polymers and biopolymers, peptides, enzymes, aptamers, antibodies, as well as with simple molecules bearing free functional groups (such as amines, carboxylates, thiols, etc.) allowing further MSN functionalization [17,18,19]. On the other hand, the inner MSN surface functionalization is crucial because it permits the anchoring/encapsulation of a wide range of functional molecules and biomolecules, complexes and nanoparticles bringing additional physical and chemical properties [20].

Among various investigated MSN, relatively recently developed dendritic or stellate-like shaped nanoparticles have attracted a particular interest. They present several advantages in comparison with their more conventional MSN counterparts: (*i*) large tunable porosity with pores size ranging from ca. 10–20 nm, permitting loading of functional species with a size larger than 5 nm (such as complexes, biomolecules, small nanoparticles, or enzymes) relevant for biomedical applications, catalysis and others, (*ii*) tunable and relatively small particle size (lower than 200 nm), (*iii*) unique dendritic pore structure, (*iv*) and a special center-radial pore morphology allowing optimized diffusion/mass transport inside the pores [21,22]. As in the case of more conventional MSN morphologies, stellate-like silica nanoparticles can be functionalized by co-condensation and post grafting approaches depending on the targeted effect. However, the works devoted to a dual functionalization of both internal and external surfaces and investigation of the functionalization impact on the nanoparticles morphology are relatively scarce [23]. Indeed, to our knowledge, there is not yet any report on the change of morphology from stellate to raspberry upon the use of an organosilane co-condensed with the tetraethyl orthosilicate (TEOS) silica precursor.

Among different applications of stellate-like MSN, their employment as a platform in the aim to protect and support catalytically active species and improve their activity has attracted a particular interest. In fact, the open porosity of the dendritic nanoparticles allows the incorporation into the pores of various large species presenting promising catalytic properties, such as enzymes, complexes, and small nanoparticles, while keeping a great diffusion of the active molecules [16,21,24]. For instance, small PdO nanoparticles for the catalysis of methane combustion [25], Pd nanoparticles for the Suzuki coupling reactions [26], and Ru nanoparticles for the hydrogenolysis of alkanes have been incorporated inside the pores [27]. Molecular catalysts were also immobilized, though the number of works on this topic is relatively scarce. We can cite the encapsulation of a tantalum hydride complex for the hydrometathesis reaction of olefins [28], heteropolyacids for the dehydration of n-butanol to di-n-butyl-ether [29] or amine functions for Knoevenagel condensation and transesterification reaction [30]. It is noteworthy that stellate-like MSN incorporating either nanoparticles [27] or molecular immobilized catalysts [29,30] demonstrated enhanced catalytic activity compared to SBA-15 or MCM-41 supports, due to their above-mentioned specific advantages. The advantage of raspberry-like morphologies has yet to be proved.

The bio-inspired [{Mn(bpy)(H_2_O)}(µ-2-MeC_6_H_4_COO)_2_(µ-O){Mn(bpy)(NO_3_)}]NO_3_ complex (denoted in the text afterwards as Mn-cat) presents an interesting catalase-like activity, which could be highly promising for the regulation of oxidative stress [31]. However, as with many bioinspired complexes, its poor solubility in water and its catalytic activity possible only in organic solvents prevents its use for biological applications. In order to overcome this problem, it has previously been encapsulated in bulk functionalized mesoporous silica MCM-41 and used as a catalyst for the decomposition of hydrogen peroxide [32]. However, the employment of bulk silica is highly limited, and MSN have already demonstrated their better accessibility for reactants, a more uniform distribution in the reaction media because of their colloidal nature, and therefore better performance. With the aim of using the full catalytic capacities of Mn-cat in aqueous medium, we develop in this work an optimized synthetic strategy [33] for the dual internal–external surface functionalization of MSN by two distinct organic functions: (*i*) 1,4-pyridine preferentially anchored on the internal surface used for the immobilization of Mn-cat [23], and (*ii*) sulfobetaine zwitterion (SBS) preferentially grafted on the external surface to ensure high colloidal stability of the nano-objects [34]. The Mn-cat complex was then immobilized inside the MSN porosity thanks to the presence of large open pores and the pyridine moieties, in order to obtain new hybrid nano-objects, which can be used as an efficient nanocatalyst. A particular emphasis is given on the investigation of the functionalization on the morphology of the nanoparticles, on the evaluation of their colloidal stability, and their catalytic activity for the dismutation of hydrogen peroxide in water.

## 2. Materials and Methods

The Mn-cat complex was synthesized according to the protocol described by Fernandez et al. in 2007 [31]. TEOS was purchased from abcr (Karlsruhe, Germany); hexadecyltrimethylammonium p-toluenesulfonate (CTATos) was purchase from Merck (Darmstadt, Germany); 2-(4-pyridylethyl)triethoxysilane (1,4-pyr) was purchased from fluorochem (Hadfield, UK); 3-dimethyl(3-trimethoxysilyl)propyl)-ammonio)propane-1-sulfonate (SBS) was purchased from Gelest (Morrisville, NC, USA); Mn(NO_3_)_2_·4H_2_O was purchased from Alfa-Aesar (Ward Hill, MA, USA); 2,2-bipyridine was purchased from TCI Europe (Zwijndrecht, Belgium); NBu_4_Br was purchased from Fluorochem (Hadfield, UK); *o*-toluic acid, KMnO_4_, ammonium nitrate and triethanolamine (TEAH_3_) were purchased from Sigma-Aldrich (Steinheim, Germany).

### 2.1. Synthsis of the Mesoporous Silica Nanoparticles

The mesoporous silica nanoparticles were synthesized by using a previously reported method with slight modifications [33].

**Pure silica nanoparticles:** 1.91 g of CTATos and 0.39 g of TEAH_3_ were dissolved in 100 mL of ultrapure water and heated at 80 °C for one hour under magnetic stirring at 500 rpm. A total of 70 mmol of TEOS was then added into the solution and which was vigorously stirred for 2 h at 80 °C. At the end of the reaction, the solution was cooled down at room temperature and the nanoparticles were collected by centrifugation at 20 krpm for 10 min. The surfactant CTATos was extracted by three washings of 30 min each in an ethanolic solution of ammonium nitrate at 6 gL^−1^ using an ultrasonic bath. The nanoparticles were then washed three times with water and one time with ethanol and dried overnight at 80 °C in an oven. 

**MSN.** Infrared (IR): υ(Si−O−Si) = 980–1250 cm^−1^ (SiO_2_), δ(O–H) = 1608 cm^−1^ (crystallized water); Thermogravimetric analysis (TGA): Estimated proportion of Si from residual mass at 1100 °C: 45.6%; Elemental analysis found (%): C(1.86), H(1.27), N(0.11); Estimated formula (M): (SiO_2_)_1_(CTA)_0.005_ (61.4 g mol^−1^); Nitrogen sorption: S_BET_ = 461 m^2^ g^−1^; d_TEM_ = 100.4 nm. 

**Hybrid silica nanoparticles with pyridine:** 1.91 g of CTATos and 0.39 g of TEAH_3_ were dissolved in 100 mL of ultrapure water and heated at 80 °C for one hour under magnetic stirring at 500 rpm. The corresponding amount of 1,4-pyr (Table 1) and TEOS (70 mmol of Si) were then added into the solution and vigorously stirred for 2 h at 80 °C. At the end of reaction, the solution was cooled down at room temperature and the nanoparticles were collected by centrifugation at 20 krpm for 10 min. The surfactant CTATos was extracted by three washings of 30 min each in an ethanolic solution of ammonium nitrate at 6 gL^−1^ using an ultrasonic bath. The nanoparticles were then washed three times with water and one time with ethanol and dried overnight at 80 °C in an oven.

**MSN-1a.** IR: υ(Si−O−Si) = 980–1250 cm^−1^ (SiO_2_), υ(C=C) = 1575 cm^−1^ (pyridine), υ(C=N) = 1610 cm^−1^ (pyridine), δ(O–H) = 1630 cm^−1^ (crystallized water); TGA: Estimated proportion of Si from residual mass at 1100 °C: 45.0%; Elemental analysis found (%): C(2.89), H(1.27), N(0.29); Estimated formula (M): (SiO_2_)_1_(CTA)_0.005_(1,4-pyr)_0.008_ (63.2 gmol^−1^); Nitrogen sorption: S_BET_ = 480 m^2^g^−1^; d_TEM_ = 79.3 nm.

**MSN-1b.** IR: υ(Si−O−Si) = 980–1250 cm^−1^ (SiO_2_), υ(C=C) = 1575 cm^−1^ (pyridine), υ(C=N) = 1610 cm^−1^ (pyridine), δ(O–H) = 1630 cm^−1^ (crystallized water); TGA: Estimated proportion of Si from residual mass at 1100 °C: 42.4%; Elemental analysis found (%): C(7.35), H(2.01), N(1.06); Estimated formula (M): (SiO_2_)_1_(CTA)_0.006_(1,4-pyr)_0.047_ (72.2 g mol^−1^); Nitrogen sorption: S_BET_ = 442 m^2^ g^−1^; d_TEM_ = 63.8 nm; δ^13^C (ss-NMR): 12.3 ppm (3C), 123.7 ppm (1C), 148.2 ppm (1C), 154.7 ppm (1C); Solid state nuclear magnetic resonance (ss-NMR): δ^29^Si (ss-NMR): −58 ppm (T_2_), −69 ppm (T_3_), −91 ppm(Q_1_), −102 ppm(Q_2_), −110 ppm(Q_3_). 

**MSN-1c.** IR: υ(Si−O−Si) = 980–1250 cm^−1^ (SiO_2_), υ(C=C) = 1575 cm^−1^ (pyridine), υ(C=N) = 1610 cm^−1^ (pyridine), δ(O–H) = 1630 cm^−1^ (crystallized water); TGA: Estimated proportion of Si from residual mass at 1100 °C: 40.0%; Elemental analysis found (%): C(12.01), H(2.88), N(1.78); Estimated formula (M): (SiO_2_)_1_(CTA)_0.007_(1,4-pyr)_0.09_ (82.1 g mol^−1^); Nitrogen sorption: S_BET_ = 592 m^2^ g^−1^; d_TEM_ = 63.7 nm. 

**Silica nanoparticles functionalized with a zwitterion on the surface:** 1.91 CTATos and 0.39 g of TEAH_3_ were dissolved in 100 mL of ultrapure water and heated at 80 °C for one hour under magnetic stirring at 500 rpm. A total of 70 mmol of TEOS was then added into the solution, which was vigorously stirred for 2 h at 80 °C. For functionalization with SBS, 1 g was added after 2 h and stirred at 80 °C overnight. At the end of reaction, the solution was cooled down at room temperature and the nanoparticles were collected by centrifugation at 20 krpm for 10 min. The surfactant CTATos was extracted by three washings of 30 min each in an ethanolic solution of ammonium nitrate at 6 gL^−1^ using an ultrasonic bath. The nanoparticles were then washed three times with water and one time with ethanol and dried overnight at 80 °C in an oven.

**MSN-2.** IR: υ(Si−O−Si) = 980–1250 cm^−1^ (SiO_2_), υ(C−N) = 1485 cm^−1^ (SBS), δ(O–H) = 1608 cm^−1^ (crystallized water); TGA: Estimated proportion of Si from residual mass at 1100 °C: 41.7%; Elemental analysis found (%): C(7.41), H(2.09), N(0.92), S(2.19); Energy dispersive spectroscopy (EDS): Si/S = 97.2/2.8; Estimated formula (M): (SiO_2_)_1_(CTA)_0.005_(SBS)_0.03_ (70.8 g mol^−1^); Nitrogen sorption: S_BET_ = 345 m^2^ g^−1^; d_TEM_ = 102.9 nm; δ^13^C (ss-NMR): 17.6 ppm (3C), 49.0 ppm (1C), 59.0 ppm (3C); δ^29^Si (ss-NMR): −69 ppm (T_3_), −91 ppm(Q_1_), −102 ppm(Q_2_), −110 ppm(Q_3_).

**Hybrid nanoparticles presenting pyridine and zwitterionic functions:** 1.91 g of CTATos and 0.39 g of TEAH_3_ were dissolved in 100 mL of ultrapure water and heated at 80 °C for one hour under magnetic stirring at 500 rpm. The corresponding amount of 1,4-pyr (Table 1) and TEOS (70 mmol of Si) were then added into the solution and vigorously stirred for 2 h at 80 °C. For functionalization with SBS, 1 g SBS was added after 2 h and stirred at 80 °C overnight. At the end of reaction, the solution was cooled down at room temperature and the nanoparticles were collected by centrifugation at 20 krpm for 10 min. The surfactant CTATos was extracted by three washings of 30 min each in an ethanolic solution of ammonium nitrate at 6 gL^−1^ using an ultrasonic bath. The nanoparticles were then washed three times with water and one time with ethanol and dried overnight at 80 °C in an oven.

**MSN-3a.** IR: υ(Si−O−Si) = 980–1250 cm^−1^ (SiO_2_), υ(C−N) = 1485 cm^−1^ (SBS), υ(C=C) = 1575 cm^−1^ (pyridine), υ(C=N) = 1610 cm^−1^ (pyridine), δ(O–H) = 1630 cm^−1^ (crystallized water); TGA: Estimated proportion of Si from residual mass at 1100 °C: 40.8%; Elemental analysis found (%): C(7.86), H(2.36), N(1.08), S(1.37); EDS: Si/S = 97.2/2.8; Estimated formula (M): (SiO_2_)_1_(CTA)_0.005_(1,4-pyr)_0.017_(SBS)_0.03_ (74.5 g mol^−1^); Nitrogen sorption: S_BET_ = 360 m^2^ g^−1^; d_TEM_ = 81.3 nm.

**MSN-3b.** IR: υ(Si−O−Si) = 980–1250 cm^−1^ (SiO_2_), υ(C−N) = 1485 cm^−1^ (SBS), υ(C=C) = 1575 cm^−1^ (pyridine), υ(C=N) = 1610 cm^−1^ (pyridine), δ(O–H) = 1630 cm^−1^ (crystallized water); TGA: Estimated proportion of Si from residual mass at 1100 °C: 40.1%; Elemental analysis found (%): C(9.41), H(2.13), N(1.46), S(1.21); EDS: Si/S = 97.4/2.6; Estimated formula (M): (SiO_2_)_1_(CTA)_0.001_(1,4-pyr)_0.048_(SBS)_0.028_ (79.7 g mol^−1^); Nitrogen sorption: S_BET_ = 423 m^2^ g^−1^; d_TEM_ = 66.8 nm. 

**MSN-3c.** IR: υ(Si−O−Si) = 980–1250 cm^−1^ (SiO_2_), υ(C−N) = 1485 cm^−1^ (SBS), υ(C=C) = 1575 cm^−1^ (pyridine), υ(C=N) = 1610 cm^−1^ (pyridine), δ(O–H) = 1630 cm^−1^ (crystallized water); TGA: Estimated proportion of Si from residual mass at 1100 °C: 38.0%; EDS: Si/S = 97.5/2.5; Elemental analysis found (%): C(13.09), H(3.00), N(2.09), S(1.05); Estimated formula (M): (SiO_2_)_1_(CTA)_0.002_(1,4-pyr)_0.096_(SBS)0.027 (74.5 g mol^−1^); Nitrogen sorption: S_BET_ = 400 m^2^ g^−1^; d_TEM_ = 72.7 nm; δ^13^C (ss-NMR): 12.3 ppm (3C), 17.6 ppm (3C), 49.0 ppm (1C), 59.0 ppm (3C), 123.7 ppm (1C), 148.2 ppm (1C), 154.7 ppm (1C).

### 2.2. Encapsulation of the Dinuclear Manganese Complex

**MSN-4a** and **MSN-4b**: 3 mg and 15 mg of Mn-cat for **MSN-4a** and **MSN-4b**, respectively, and 30 mg of **MSN-3c** were suspended in 25 mL of water and stirred overnight at 80 °C. A brown solid was then collected by centrifugation at 20 krpm for 10 min and the supernatant was colorless. The brown powder collected was then washed three times in CH_3_CN to remove extra Mn-cat and finally dried overnight in an oven at 80 °C.

**MSN-4a.** EDS: Si/Mn = 98.4/1.6; Estimated formula (M): (SiO_2_)_1_(CTA)_0.001_(1,4-pyr)_0.096_(SBS)_0.027_(Mn-cat)_0.0096_ (98.4 g mol^−1^); Nitrogen sorption: S_BET_ = 235 m^2^ g^−1^; d_TEM_ = 72.5 nm.

**MSN-4b.** EDS: Si/Mn = 92.2/7.8; Estimated formula (M): (SiO_2_)_1_(CTA)_0.001_(1,4-pyr)_0.096_(SBS)_0.027_(Mn-cat)_0.053_ (136.1 g mol^−1^); Nitrogen sorption: S_BET_ = 167 m^2^ g^−1^; d_TEM_ = 71.1 nm.

### 2.3. Nanoparticle’s Characterization

Transmission Electron Microscopy (TEM) images were performed at 100 kV (JEOL 1200 EXII, Tokyo, Japan). Samples for TEM measurements were deposited from suspensions on copper grids and allowed to dry before observation. The size distribution histograms were determined using enlarged TEM micrographs taken at magnification of 100 K on a statistical sample of ca. 100 NPs. Scanning electron microscopy and EDS were performed on a FEI Quanta FEG 200 instrument (Hillsboro, OR, USA). The powders were deposited on an adhesive carbon film and analyzed under high vacuum. The quantification of the heavy elements was carried out with the INCA software, with a dwell time of 3 µs. High resolution transmission electron microscopy (HRTEM) and STEM-EDS measurements were performed on a JEOL 2200 FS instrument (Tokyo, Japan). Zeta potential and hydrodynamic diameter measurements were recorded on Malvern nanoseries (Malvern, UK), Zetasizer NanoZS (Model ZEN3600) in a DTS1060C Zetacell (for the zeta potential) in water at 25 °C with an equilibration time of 120 s with automatic measurement and data were treated by Zetasizer software using a Smoluchowski model. TGA was performed using a Netsch STA 409 PC analyzer in the temperature range 20–1100 °C at a heating speed of 5 °C min^−1^ under air. The nanoparticles’ composition was determined by elemental analysis (C, H, N, S) and the amount of Si was calculated based on the TGA results by considering the resulting mass at 1100 °C as pure SiO_2_. Elemental analyses were performed with an Elementar Vario Micro Cube analyzer (Billerica, MA, USA). Powders were pyrolyzed at 1150 °C and then reduced at 850 °C over hot copper. Gases were separated by gas chromatography. Infrared spectra were recorded by attenuated total reflectance on a Perkin Elmer Spectrum (Waltham, MA, USA) Two spectrophotometer with four acquisitions at a resolution of 4 cm^−1^. Nitrogen adsorption and desorption isotherms at 77 K were measured using a TriStar 3000 (Norcross, GA, USA) (V6.06 A). Prior to the sorption experiment, the samples were dried under vacuum at 80 °C for 12 h. The specific surface area (S_BET_) was calculated according to the Brunauer–Emmett–Teller (BET) method. Pore size distribution was obtained from the desorption branch of the nitrogen isotherm using the Barrett–Joyner–Halenda (BJH) equation. All solid-state NMR experiments were performed on a Varian VNMRS 300 MHz (7.05 T) NMR spectrometer (Santa Clara, CA, USA). A 3.2 mm Varian T3 HXY magic angle spinning (MAS) probe was used for ^13^C and ^29^Si experiments, at operating frequencies of 599.82 and 564.33 MHz, respectively. ^1^H–^13^C CPMAS NMR spectra were recorded spinning at 5 kHz, with a contact time of 2.5 ms and 300 kHz, ^1^H decoupling during acquisition. A total of 15,000 scans were recorded for each measurement. ^13^C chemical shifts were referenced externally to adamantane (used as a secondary reference), the high frequency peak being set to 38.5 ppm.

### 2.4. Resistance to Protein-Induced Aggregation and Colloidal Stability

The nanoparticles’ resistance against protein adsorption and its colloidal stability were analyzed by following the hydrodynamic diameter of the nanoparticles by DLS. Nanoparticles were suspended at 1 mg mL^−1^ in 10 mL of water or of a mix of fetal bovine serum (FBS) and Dulbecco’s modified Eagle’s medium (DMEM) high glucose. The hydrodynamic diameter of a 2 mL solution in a cell was followed over a few days for **MSN-3c**. 

### 2.5. Catalytic Tests

For each test, the number of nanoparticles was chosen and calculated to keep the same amount of Mn-cat (3.9 µmol) in the solution. In a closed flask of 25 mL containing a suspension of nanoparticles in CH_3_CN or water, 0.2 mL of hydrogen peroxide 30% (H_2_O_2_ without inhibitors) were added and the volume of O_2_ produced by the dismutation of H_2_O_2_ was tracked over time. The reaction advancement was calculated based on a theoretical volume of 24 mL considering the dioxygen as a perfect gas in standard conditions (1 atm., 25 °C). For comparison, the same experiments were done using the same amount of Mn-cat dissolved in CH_3_CN and water. To address the catalyst resistance/recyclability, the measurements were restarted in situ with the same materials several times.

## 3. Results and Discussion

### 3.1. Synthesis and Characterisations

The design of hybrid nanocatalysts was performed through encapsulation of the bio-inspired [{Mn(bpy)(H_2_O)}(µ-2-MeC_6_H_4_COO)_2_(µ-O){Mn(bpy)(NO_3_)}]NO_3_ complex inside the porosity of dually functionalized mesostructured silica by using the two step procedure depicted in Figure 1. In the first step, the functionalized MSN nanoparticles (**MSN-3a–c**) containing 1,4-pyridine moieties anchored inside the pores and sulfobetaine zwitterion on the nanoparticles surface were synthesized by a sol–gel process using CTATos as surfactant and TEAH_3_ as basic catalyst. The functionalization of the MSNs was performed in situ, first by the co-condensation procedure during the sol–gel reaction by adding 1,4-pyridine siloxane and TEOS for its preferential grafting within the nanoparticles, and second by post-functionalization with SBS to promote its preferential grafting at the nanoparticles’ periphery as the pores were obstructed by the surfactant. The latter was then removed by further thorough washings to liberate the porosity and to allow accessibility to the 1,4-pyridine, even though it should be noted that some traces of residual CTA could always be observed. In order to evaluate the impact of 1,4-pyridine moiety on the morphology of nanoparticles, different amounts were anchored inside the porosity leading to the nanoparticles **MSN-3a**, **MSN-3b** and **MSN-3c** (see Table 1 for the initial number of reagents and Table 2 for characteristics).

Secondly, the bioinspired [{Mn(bpy)(H_2_O)}(µ-2-MeC_6_H_4_COO)_2_(µ-O){Mn(bpy)(NO_3_)}]NO_3_ (Mn-cat) compound was then incorporated into the silica pores of the nanoparticles **MSN-3c** containing the largest amount of the grafted 1,4-pyridine by its impregnation in water. The manganese complex is poorly soluble in water, but the internal structure of the hybrid silica presents a hydrophobic environment thanks to the pyridine functionalization, which is favorable for the complex incorporation into the nanoparticles. During this reaction, a notable change of color of the silica powder from white to brownish concomitantly with a complete discoloration of the supernatant solution was consistent with a quantitative incorporation of the Mn-cat complexes inside the nanoparticle porosity. The relation between the catalytic activity and the amount of inserted Mn-cat was investigated comparing samples **MSN-4a**,**b** containing different Mn-cat loadings (composition from EDS, elemental analysis and TGA, see experimental, Appendix A, Table 2). Relative to the initial concentration, the Mn-cat retention yields were 92.7% and 98.1%, corresponding to 0.0096 and 0.053 Mn-cat per SiO_2_ unit in **MSN-4a** and **MSN-4b**, respectively. Relative to the pyridine moieties eventually available to coordinate Mn-cat, it corresponds to Mn-cat/pyridine ratio of 0.10 and 0.55 for **MSN-4a** and **MSN-4b**, respectively.

In order to investigate the impact of the inner and external surface functionalization with 1,4-pyridine and zwitterion, non-functionalized MSN (**MSN**) and mono-functionalized MSN, either with 1,4-pyridine alone with different amounts (**MSN-1a**–**c**) or SBS alone, (**MSN-2**) were also prepared and characterized for comparison.

As described by Zhang et al. [33], the synthesis of pristine non-functionalized stellate-like silica **MSN** leads to the formation of nanoparticles presenting a size of ca. 100 nm with an open morphology (Figure 1A). However, the functionalization of their inner surface with pyridine siloxane by using the in situ approach (sample **MSN-1c**) led to two notable modifications (Figure 1B): (*i*) a slight change of the nanoparticles’ shape from stellate to an intermediate stellate-raspberry-like one, (*ii*) and a size decrease from 100.4 nm to 63.7 nm (Table 2). On the other hand, the addition of the SBS at the end of the nanoparticles’ growth with or without the pyridine moieties (samples **MSN-3a–c** or **MSN-2**, respectively) has a minor impact on their size (100.4 nm vs. 102.9 nm without pyridine or 63.7 nm vs. 72.7 nm with pyridine, see Table 1) or their morphology (Figure 1C and Appendix A). Such effects are induced by the addition of the pyridine siloxane into the MSN and the absence of any major influence of the zwitterion siloxane is consistent with a preferential location of each organic function in the core or at the external corona of the nanoparticles, respectively.

Finally, the post-synthetic incorporation of Mn-cat into the silica porosity (sample **MSN-4a** and **MSN-4b**) did not bring any changes in the morphology of the nanoparticles (Figure 1D, Figure 2 and Appendix A). The nanoparticles keep both their size (Table 1) and their stellate-raspberry-like shape independently on the amount of incorporated Mn-cat.

Nitrogen sorption measurements were performed on the different nanomaterials obtained (Figure 2 and Appendix A) to assess the impact of each step of functionalization on both surface area and pore size. As observed by TEM, non-functionalized **MSN** and only SBS-functionalized **MSN-2** possess a typical stellate-like morphology with an open porosity presenting pore sizes of 19 nm and 17 nm and high specific surfaces of 547 m^2^ g^−1^ and 480 m^2^ g^−1^, respectively. The slightly lowest accessible surface in the latter may be attributed to the presence of the sulfobetaine zwitterion at the periphery, which may partly obstruct the pore entry. Comparatively, introducing 1,4-pyridine siloxane during the sol–gel reaction (**MSN-1c**) leads to an obvious change of morphology from stellate-like to intermediate stellate-raspberry-like, together with a slightly more pronounced decrease in surface area. At low level of addition (0.008 1,4pyr/TEOS) the surface area falls down to 480 m^2^ g^−1^ and even lower to 442 and 407 m^2^ g^−1^ at a higher level of addition (0.047, 0.090 1,4pyr/TEOS). Comparing **MSN-1** series to **MSN-3** series of nanoparticles suggests that the effect is additive moving from mono to dual functionalization (Figure 3, Appendix A). Morphology changes and loss of surface area are consistent with the specific location of the 1,4-pyridine function in the core of the nanoparticles and the sulfobetaine zwitterion at their more open periphery. Even if the pores are less open than in pristine stellate nanoparticles [33], the pore size distribution of the present dual-functionalized nanoparticles remains large and centered at a larger value than a single micelle would template (ca. 7 nm instead of 3 nm) (Appendix A) [33,35,36]. Indeed, a close look to the external contour of the nanoparticle reveals a marked roughness of 10 to 15 nm, close to that of stellate nanoparticles, although in absence of any radial aspect expected for this morphology, placing these nanoparticles in an intermediate state between stellate and raspberry.

The encapsulation of the Mn-cat complex leads to further decrease in the specific surface area to 235 m^2^ g^−1^ for **MSN-4a** and 166 m^2^ g^−1^ for **MSN-4b** (Figure 3 and Appendix A, Table 2). Such results are coherent with an increased clogging of the pores consequent to the incorporation of the complex within the porosity where pyridine moieties are available. Moreover, **MSN-4b** presenting higher amount of the incorporated complex exhibits a much lower specific surface area. 

IR spectroscopy allowed us to confirm the grafting of the 1,4-pyridine and sulfobetaine zwitterion moieties into the silica nanoparticles (Appendix A). In addition to the C−H vibrational band between 2900–3000 cm^−1^ we can identify various bands from the pyridine function (υ(C=N) = 1610 cm^−1^, υ(C=C): 1575 cm^−1^, δ(Si–O–C): 1420 cm^−1^) and the zwitterion function (υ(C−NH_3_): 1485 cm^−1^, δ(Si–O–C): 1420 cm^−1^) (Appendix A) [37]. Furthermore, a new band at 1440 cm^−1^ specific to the bipyridine function arising from the Mn-cat complex can be identified on the IR spectrum of **MSN-4b** confirming therefore the incorporation of the latter (Appendix A) [32]. On the other hand, the disappearance of the strong stretching vibration of the NO_3_^−^ group on the IR spectrum of **MSN-4b**, which is clearly visible at 1375 cm^−1^ of the spectrum of the initial complex Mn-cat (Appendix A), suggests the probable coordination of Mn-cat to the pyridine functions of silica during immobilization and the loss of the labile NO_3_^−^ [31].

The grafting of the pyridine siloxane (**MSN-1b**) and SBS zwitterion siloxane (**MSN-2**) were also confirmed by using ^29^Si solid-state NMR, which shows the presence of the T_2_ and T_3_ signals at −59 ppm and −69 ppm, respectively (Appendix A). The same samples were characterized by ^13^C solid state NMR. In both samples, we observe the peaks at 23, 30, 33 and 54 ppm due to residual CTA. For **MSN-1b**, we can also clearly identify the unshielded ^13^C signals of the pyridine aromatic carbons at 124, 148 and 155 ppm and the methylene alkane chain carbons as a broad peak around 10 ppm (Appendix A). On the ^13^C NMR spectrum of **MSN-2** we can identify the carbon signals in α position of the nitrogen of the ammonium as a broad peak at 61 ppm, the carbon signal in α position of the sulfonate group at 49 ppm and the other methylene carbons of the alkane chain as a broad peak at 12 ppm. The grafting of both 1,4-pyridine and zwitterion in sample **MSN-3c** was also confirmed by ^13^C NMR. The peak characteristic for each function is clearly visible in the spectrum (Appendix A). The NMR characterizations of **MSN-4a** and **MSN-4b** have not been done due to the paramagnetic behavior of the Mn-cat complex at room temperature.

Further, the obtained nanocatalysts have been characterized by STEM/EDS analysis, which allows visualizing the homogeneity of atomic distribution of sulfur from the sulfobetaine zwitterion moieties and manganese atoms from the Mn-cat complex. Figure 4 demonstrates the homogeneous distribution of sulfur and the presence of manganese atoms confirming the encapsulation of the Mn-cat complex for **MSN-4b**.

The zeta potential and the colloidal stability were investigated for the non-functionalized, mono- and dual-functionalized nanoparticles (Appendix A) [34]. Indeed, the functions affect the zeta potential that obviously shift to a higher pH. The effect is well pronounced in the presence of 1,4-pyr functions with the zero-point charge moving from a pH of ca. 2.5 to 3.9 for **MSN-1a** and 5.1 for **MSN-1c,** and rather modest for the zwitterionic SBS function, ca. 3.2 for **MSN-2** (Appendix A). Apparently, the effect of each function is additive as in the dual-functionalized nanoparticles **MSN-3c**, the zero-point charge moves to a slightly higher pH, ca. 5.5. In comparison to non-porous silica nanoparticles grafted with a zwitterion, the zeta potential remains zero. This difference can be explained, in part, by the lower amount of grafted zwitterion in our work (0.69 μmol m^−2^ instead of 1.7 μmol m^−2^ in ref. [34]). The open porosity is likely the second reason as it allows the exposition of some 1,4-pyr functions and certainly also some silanol groups to the environment, preventing the achievement of a zero zeta potential.

A major effect of the zwitterion anchoring on the nanoparticles surface was observed in the colloidal stability. Indeed, after drying the nanoparticles, the non-functionalized **MSN** or the 1,4-pyr functionalized **MSN-1a**–**c** cannot be dispersed in water at 1 mgmL^−1^ even with an ultrasound treatment, while nanoparticles **MSN-3c** with 10% pyridine and zwitterion were easily dispersed in water and in a biological medium (DMEM high glucose and FBS 50/50) and showed colloidal stability over a few days (Figure 5). Thus, the little amount of zwitterion grafted on the external surface (2.6%, Table 2) was enough to create a water barrier against adsorption and aggregation, even though silanol groups from the porosity may also be exposed to the environment of the MSN [38]. Similar colloidal stability is observed for hybrid nanoparticles **MSN4a** and **MSN4b** containing the inserted Mn-cat complex. 

### 3.2. H_2_O_2_ Catalytic Dismutation Reaction

The catalase activity was determined by measuring the advancement of the dismutation reaction of hydrogen peroxide (2H_2_O_2_ → 2H_2_O + O_2_) in the presence of a suspension of **MSN-4a** or **MSN-4b** nanoparticles and compared with the activity of free-Mn-cat complex in water or in CH_3_CN. To allow direct comparison of the respective catalytic activities, the same amount of free- and inserted Mn-cat complex in the nanoparticles was used for the catalytic tests to keep a comparable amount of the active species. The results are shown in Figure 6 and the catalytic activity data are summarized in Appendix A. In acetonitrile, the hydrogen peroxide conversion after 1 and 5 min is of 44% and 89% for **MSN-4a**, 27% and 77% for **MSN-4b** and 33% and 73% for free-Mn-cat, respectively. Notably, **MSN4a** contains 5.5 times less immobilized Mn-cat complex than **MSN-4b** and shows a greater catalytic activity, while being normalized relative to the active catalytic centers. That may be attributed to the increased pore clogging in the latter, as shown from N_2_-sorption experiments and thus to the lower accessibility of reactive H_2_O_2_ to the catalytic centers. After 20 min, 100% conversion is obtained for **MSN4a**, **b** and 95% for free-Mn-cat. In water, a large difference between the free-Mn-cat and the inserted one into the silica pores in **MSN-4a**, **b** is obtained. Indeed, while nanoparticles show excellent catalytic activity with hydrogen peroxide with the 63% and 100% of conversion for **MSN-4a** and 58% and 100% for **MSN-4b** after 1 and 5 min, respectively, the free-Mn-cat compound shows almost no activity with 0.3% and 1% of conversion after 1 and 5 min. The extremely low activity of the latter is explained by its poor solubility in water. Such a difference emphasizes the interest in Mn-cat encapsulation in MSN to preserve its catalytic activity for further biological experiments [32].

The comparison of the results obtained in our system with the ones of the Mn-cat complex immobilized into a bulk mesoporous LUS-type silica normalized by the ratio [Mn-cat]/[H_2_O_2_] of the catalytic species [32] indicates that the catalytic activity of both systems is similar, as expected for the systems with full accessibility of catalytic species in the pores. As an example, with a fixed Mn-cat unit/H_2_O_2_ ratio of 6.80 × 10^−4^, the Mn-cat complex immobilized into a bulk mesoporous LUS-type silica gave 19% of the dismutation of H_2_O_2_ in water after 1 min. When using **MSN-4a** and **MSN-4b** with a fixed Mn-cat unit/H_2_O_2_ ratio of 1.99 × 10^−3^, the dismutation of H_2_O_2_ in water after 1 min is 63% and 58%, respectively. If both activities are thus similar, the total material amount of **MSN-4b** nanocatalyst needed is five times lower than the total material amount for Mn-cat/LUS-type mesoporous bulk silica. This result confirms the interest in the elaborated strategy to design an efficient nanocatalyst working in water with an increased amount of the inserted catalytically active species (Mn-cat complex), while preserving its full activity and integrity. 

Even more interesting and unexpected is the fact that the catalytic activity of **MSN-4a**,**b** is greater in water than in acetonitrile. Indeed, after 5 min, 100% conversion is obtained for **MSN-4a**,**b** in water, while it is 89% and 77% for **MSN-4a** and **MSN4b**, respectively, in acetonitrile. This can be attributed to the colloidal behavior of the nanoparticles: the nanoparticles in CH_3_CN aggregate during the catalytic process which may limit access to the active catalytic centers; whereas, in water, thanks to the presence of the zwitterion, the nanoparticles show a great colloidal stability and excellent dispersibility. This increases the accessibility to its inner hydrophobic environment and thus the accessibility to the complex and its catalytic activity. Such results underline the interest of the dual functionalization developed here for the immobilization of a molecular catalyst for reactions in an aqueous medium.

To check the robustness of the nanomaterial, three successive cycles consisting of additions of the same volume of H_2_O_2_ in the same solution were performed (Figure 7). **MSN-4a** showing superior catalytic activity was chosen for these experiments. The results are shown in Figure 7 with the conversion of H_2_O_2_ in % and turnover number (TON) over time (defined as moles of O_2_ evolved per mole of catalyst); the latter provides the efficiency of the catalyst. The initial rate slopes were determined in order to compare the efficiency of the system during its first use and the two following ones. There was a significant difference in the initial rate slopes with an important drop between the first and second runs with values of 0.62 s^−1^ and 0.14 s^−1^, while between the second and third runs, the initial rate slopes were comparable with 0.12 s^−1^ for the latter. Furthermore, a slight delay can be observed on the second and third runs, while not present for the first run (Appendix A). Those results may suggest that at the end of the first run, the remaining state of the catalyst was different from the initial state, which is the reason for the induction time of the second and third runs [32]. Furthermore, there is an additional decrease in the decomposition speed for the second and third cycles. Very similar results were observed with the complex immobilized within bulk silica, with a slight loss of speed and a slight delay at the beginning [32]. Such a delay was explained by the remainder of the complex being in a different state after the first run, and the loss of catalytic activity was explained by changes in the pH after the addition of hydrogen peroxide. Indeed, the catalase activity is known to be greatly affected by the acidification of the media after the additions of H_2_O_2_.

## 4. Conclusions

In summary, in this work, we successfully designed a new efficient hybrid nanocatalyst through the encapsulation of the bio-inspired [{Mn(bpy)(H_2_O)}(µ-2-MeC_6_H_4_COO)_2_(µ-O){Mn(bpy)(NO_3_)}]NO_3_ complex inside the porosity of dually functionalized stellate-raspberry-like mesostructured silica and demonstrated its catalytic activity for the dismutation reaction of hydrogen peroxide. 

First, we developed a series of stellate-raspberry-like MSN with a size of ca. 72 nm presenting a dual internal–external functionalization with 1,4-pyridine and the sulfobetaine zwitterion. The preferential localization of the 1,4-pyridine moiety within the cavities inside the silica porosity and the preferential location of the sulfobetaine zwitterion on the nanoparticles’ surface were performed through an in situ approach implying the growth of the silica stellate-raspberry-like architecture in presence of the templating agent concomitant with the grafting of the 1,4-pyridine siloxane, followed by the anchoring of the sulfobetaine zwitterion just before removal of the templating agent. The success of this dual internal–external functionalization was demonstrated through a bundle of structural and textural characterization techniques and systematic comparison with the pristine and the mono-functionalized, with only 1,4-pyridine or with only the sulfobetaine zwitterion stellate-like MSN. The dual-functionalized stellate-raspberry-like MSN present a size of ca. 72 nm and large size of pores (7.6–14.5 nm), which depend on the amount of 1,4-pyridine anchored and which are suitable for the encapsulation of catalytically active species of a relatively large size.

Secondly, the as-obtained stellate-raspberry-like MSN were further used for the encapsulation of the bio-inspired [{Mn(bpy)(H_2_O)}(µ-2-MeC_6_H_4_COO)_2_(µ-O){Mn(bpy)(NO_3_)}]NO_3_ complex through the pyridine moieties generating hydrophobic pockets in the cavities of the raspberry-stellate-like MSN, auspicious for the manganese complex encapsulation. The latter presents interesting catalase-like activity, which could be promising for the regulation of the oxidative stress, but of limited use due to its poor solubility and catalytic activity in water. We showed that its immobilization within the as-prepared stellate-raspberry-like MSN allows exploiting the catalase-mimicking activity of this manganese complex through the dismutation reaction of hydrogen peroxide in water, while the manganese complex in the same conditions is inefficient. The obtained normalized catalytic activities for the Mn-cat complex immobilized into stellate-raspberry-type MSN and in the porosity of the bulk LUS-type mesoporous silica are similar, as expected for efficient porous systems offering a full accessibility of an inserted catalytic species. However, the amount of the nanocatalyst employed for the tests is much more favorable in comparison with the bulk mesoporous silica. Of note, the activity of the as-obtained nanocatalysts is even greater in water than in acetonitrile, thanks, according to our hypothesis, to the presence of the sulfobetaine zwitterion that prevents nanoparticle aggregation in water more efficiently. The robustness and efficiency of the nanocatalyst was confirmed through its successive uses in water showing that its catalytic activity is maintained with minimal activity loss after the first run. Taking into account the excellent colloidal stability of these nano-objects, this work represents a further step toward the rational design of efficient nanocatalysts for biomedical applications and can serve as a model nano-system for other catalytic systems exploiting hydrophobic complexes that need to be used in aqueous media. The developed dually functionalized stellate-raspberry-like MSN provide an appropriate environment for the catalytic operation of the sequestered complexes in water and can even make an improvement on their catalytic activity.

## Data Availability

The data presented in this study are available on request from the corresponding authors.

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
