# Peer review of "Multifunctionalized Mesostructured Silica Nanoparticles Containing Mn2 Complex for Improved Catalase-Mimicking Activity in Water"

_nanomaterials, 2022, doi:10.3390/nano12071136_

Round 1

Reviewer 1 Report

The manuscript deals with a method to functionalise silica nanoparticles for mimicking catalase activity. The authors did a considerable work to fabricate and characterise different kinds of nanoparticles and used a variety of chemical functionalisation steps. In my opinion their work demonstrate with sufficient soundness their conclusions.

I have however some minor suggestions before publication:

  1. As a general rule, all short names, however they might be thought as generally known, have to be explicitly defined at their first use within the manuscript. This is not the case for several short names within this manuscript, so I advise the authors to correct the text along my suggestion.
  2. given the large number of acronyms used by the authors, a summary of them at the end of the manuscript would greatly help the readers to keep track of them and therefore improve the readability of the manuscript
  3. The authors define part of the acronyms, but they redefine some of them multiple times within the text. Unless strictly necessary for clarity, this should be avoided
  4. Even if they define several acronyms, some of them is randomly not used within the text. The use of the acronyms has to be revisited
  5. The authors use the Mn2O short name for the main chemical compound used within this study. I suggest using a different indication for the compound, since the one chosen is a chemical formula and may induce the reader in erroneous reading of the manuscript results. See for reference:  https://pubchem.ncbi.nlm.nih.gov/compound/59475671 or https://pubchem.ncbi.nlm.nih.gov/#query=Mn2O 
  6. On line 92 reference 26 is miscited. Ref. 27 has to be cited instead.

Author Response

We have carefully taken into account the comments and suggestions of the reviewers. We have listed below in more detail our responses (in bold) to the reviewer’s comments and suggestions.

Comments and Suggestions for Authors

The manuscript deals with a method to functionalise silica nanoparticles for mimicking catalase activity. The authors did a considerable work to fabricate and characterise different kinds of nanoparticles and used a variety of chemical functionalisation steps. In my opinion their work demonstrate with sufficient soundness their conclusions.

We thank the reviewer for his(her) positive opinion on the work presented in this article.

I have however some minor suggestions before publication:

  1. As a general rule, all short names, however they might be thought as generally known, have to be explicitly defined at their first use within the manuscript. This is not the case for several short names within this manuscript, so I advise the authors to correct the text along my suggestion.

As suggested by the reviewer, all short names are now explicitly defined at their first use within the manuscript.

  1. given the large number of acronyms used by the authors, a summary of them at the end of the manuscript would greatly help the readers to keep track of them and therefore improve the readability of the manuscript

As suggested by the reviewer, a list of acronyms is now given after the conclusion.

  1. The authors define part of the acronyms, but they redefine some of them multiple times within the text. Unless strictly necessary for clarity, this should be avoided

As suggested by the reviewer, acronyms are now defined only once when used for the first time.

  1. Even if they define several acronyms, some of them is randomly not used within the text. The use of the acronyms has to be revisited

In line with the reviewer comments and the above comments, the use of acronyms was completerly revisited to meet his(her) demands.

  1. The authors use the Mn2O short name for the main chemical compound used within this study. I suggest using a different indication for the compound, since the one chosen is a chemical formula and may induce the reader in erroneous reading of the manuscript results. See for reference:  https://pubchem.ncbi.nlm.nih.gov/compound/59475671 or https://pubchem.ncbi.nlm.nih.gov/#query=Mn2O 

We agree with the reviewer on the misinterpretation that could result from the use of the Mn2O short name. Thus it was changed for Mn-cat shortname.

  1. On line 92 reference 26 is miscited. Ref. 27 has to be cited instead.

In the new version of the article after revision, references 26 and 27 are now references 31 and 32. However, we maintain reference 26 (31) mentioned line 92 because it refers to the molecular complex while reference 27 (32) refers to the complex immobilized on a silica.

Reviewer 2 Report

"The manuscript entitlet "Multifunctionalized Mesostructured Silica Nanoparticles containing Mn2 complex for improved catalase mimicking activity in water" by B. Albela, J. Larionova, Y. Guari and cowoorkers describes the succesfully designed synthesis of Multifunctionalized mesostructured silica nanoparticles containing Mn complex. The manuscript is well written and the compounds well characterized and in agreement with the literature.

Author Response

We have carefully taken into account the comments and suggestions of the reviewers. We have listed below in more detail our responses (in bold) to the reviewer’s comments and suggestions.

Comments and Suggestions for Authors

"The manuscript entitlet "Multifunctionalized Mesostructured Silica Nanoparticles containing Mncomplex for improved catalase mimicking activity in water" by B. Albela, J. Larionova, Y. Guari and cowoorkers describes the succesfully designed synthesis of Multifunctionalized mesostructured silica nanoparticles containing Mn complex. The manuscript is well written and the compounds well characterized and in agreement with the literature.

We thank the reviewer for his very good opinion of the work and results obtained in the context of this study.

Reviewer 3 Report

The authors presented an intriguing work about the functionalization of silica nanoparticles with a Mn complex to improve the catalase activity of the compound also in water solution and for medical purpose. The compound, at the end of the synthesis, took a stellate-raspberry-shape with nanometric dimensions. The synthesis and the characterization of the final and intermediate compounds were well written, documented and based on the experimental results. Several tests on H2O2 were carried out in presence of the new compounds and the results appeared very good. The article could be of interest to design and produce innovative materials and new catalysts. I suggest accepting the manuscript after minor revisions:

  • a deep revision of the references that appeared not ever in the appropriate form
  • measurement units (atm, min, etc.) should be written without the point, not in the form atm., min., etc. In the case of gL-1 and similia, I suggest that they were indicated without the point (g.L-1)
  • the “nanocatalyst” word should be written as a whole one; in the text it is present both in this form and as “nano-catalyst”
  • the acronyms should be explicated ONLY the FIRST time that they are written in the text; later, only acronyms should be used.
  • the acronym TEOS was not explicated
  • 1 line 36-37: I’d add some references about the use in catalysis, waste remediation
  • 3 line 121 and line 136: the concentration of TEOS should be indicated in the same manner
  • 10 line 390: the sign – in the nitrate group should be superscript

Author Response

We have carefully taken into account the comments and suggestions of the reviewers. We have listed below in more detail our responses (in bold) to the reviewer’s comments and suggestions.

Comments and Suggestions for Authors

The authors presented an intriguing work about the functionalization of silica nanoparticles with a Mn complex to improve the catalase activity of the compound also in water solution and for medical purpose. The compound, at the end of the synthesis, took a stellate-raspberry-shape with nanometric dimensions. The synthesis and the characterization of the final and intermediate compounds were well written, documented and based on the experimental results. Several tests on H2O2 were carried out in presence of the new compounds and the results appeared very good. The article could be of interest to design and produce innovative materials and new catalysts. I suggest accepting the manuscript after minor revisions:

We would like to thank the reviewer for his very good opinion of the results described in this article.

  • a deep revision of the references that appeared not ever in the appropriate form

We apologize for the erroneous use of an inappropriate reference style, this has been modified using the MDPI reference style as suggested by the reviewer.

  • measurement units (atm, min, etc.) should be written without the point, not in the form atm., min., etc. In the case of gL-1and similia, I suggest that they were indicated without the point (g.L-1)

As suggested by the reviewer, all dots have been removed for measurement units.

  • the “nanocatalyst” word should be written as a whole one; in the text it is present both in this form and as “nano-catalyst”

Only the word nanocatalyst now appears throughout the text.

  • the acronyms should be explicated ONLY the FIRST time that they are written in the text; later, only acronyms should be used.

Following the recommendation of the reviewer and in connection with the comment of the first reviewer, the acronyms are explained only once when they appear for the first time in the text.

  • the acronym TEOS was not explicated

The acronym TEOS is now explicated at his first appearance line 71.

  • 1 line 36-37: I’d add some references about the use in catalysis, waste remediation

As suggested by the reviewer, we have added 5 journal references concerning the use of this type of material for catalysis or waste remediation; references 2, 3, 4, 7 and 8.

  • 3 line 121 and line 136: the concentration of TEOS should be indicated in the same manner

Done

  • 10 line 390: the sign – in the nitrate group should be superscript

Done